# OpenReview forum: "Jailbreak-AudioBench: In-Depth Evaluation and Analysis of Jailbreak Threats for Large Audio Language Models"
_NeurIPS.cc/2025/Datasets_and_Benchmarks_Track — NeurIPS 2025 Datasets and Benchmarks Track poster_

### Official Review · Reviewer_hifu · 2025-07-01

**Rating:** 5
**Confidence:** 3

**Summary:**

The paper introduces Jailbreak-AudioBench, a benchmark for testing how vulnerable end-to-end Large Audio-Language Models (LALMs) are to audio-based jailbreaks. It provides a toolbox for audio editing (Emphasis, Speed, Intonation, Tone, Background Noise, Celebrity Accent , Emotion), a 157K-sample dataset, evaluations of multiple LALMs, a query-based attack method, and a simple defense strategy.

**Dataset Code Accessibility:**

Yes

**Dataset Code Comments:**

The authors provided Huggingface repository to access the dataset and Github repository to reproduce results in the paper.

**Ethical Comments:**

I think this is a strong paper that will help pave the way for research on jailbreaking LLM in the audio domain, thus I am keeping my score.

**Ethical Considerations:**

No, there are no or only very minor ethics concerns

**Final Justification:**

As mentioned, I think this is a strong paper that will help pave the way for research on jailbreaking LLMs in the audio domain. Thus I am keeping my score to accept this paper

**Limitations Weaknesses:**

Currently I am inclined to accept this paper since I don't find any major weakness. My score may change after the rebuttal period.

**Strengths Contributions:**

- The proposed query-based audio editing method is simple yet effective, significantly increasing the attack success rate across models. The paper also presents a lightweight defense strategy using prepended safety prompts. However, it remains unclear whether combining multiple types of audio edits—beyond those explored—could further amplify attack success rates.

- The paper introduces a large and diverse dataset (157K samples) with explicit and implicit jailbreak categories. However, the writing could be improved—specifically, the criteria for categorizing audio as explicit or implicit is not clearly explained, which may confuse readers trying to understand the dataset structure.

- The t-SNE visualizations of token embedding spaces offer valuable insight into model behavior, effectively illustrating how different models vary in their robustness to audio edits.

---

> ### Author Rebuttal · Authors · 2025-07-31
>
> # Response to Reviewer hifu
>
> Dear Reviewer hifu,
>
> We appreciate your thoughtful feedback on our paper. Regarding your concern, we include additional comments and results to further improve our work. Thanks for your insightful feedback and for recognizing the core strengths of our work.
>
> ## Strength (S) and Questions (Q):
>
> **S1:** We are particularly pleased that you found our query-based audio editing method to be simple but effective. Your recognition of its significant impact on increasing the Attack Success Rate (ASR) truly captures the goal of that experiment. As illustrated in Figure 5 and Figure 8 in Section 4.1 and Appendix D.1, query-based audio editing jailbreak methods can achieve ASR improvements across different open and closed source end-to-end Large Audio Language Models (LALMs).
>
> > **Q1: It remains unclear whether combining multiple types of audio edits—beyond those explored—could further amplify attack success rates.**
>
> **A1:** Thank you for this insightful feedback. We appreciate the opportunity to clarify this important aspect of our work. In our paper, we indeed explored the combination of multiple audio editing types beyond individual modifications. We first evaluated distinct audio editing types individually: Tone, Intonation, Speed, Emphasis, Background Noise, Celebrity Accent, and Emotion modifications. This baseline analysis revealed varying degrees of vulnerability across different Large Audio Language Models, as shown in Table 2 of the paper.
>
> Building upon our t-SNE and UMAP analysis in Figure 4 of the paper and Appendix Section C.3, we observed that vulnerable models maintain clear editing-based clustering throughout the Large Audio Language Model's entire architecture, where distinct clusters for different audio editing remain separated from original audio samples, explaining their high susceptibility to audio editing jailbreak. This finding suggested that combining multiple audio editing types may generate features with even greater separation from original audio samples, potentially amplifying jailbreak Attack Success Rate (ASR).
>
> Motivated by this insight, we developed the Query-based Audio Editing Method that systematically combines multiple audio editing types (Lines 204-223). As demonstrated in Figure 5 of the paper and Appendix Figure 8, this approach significantly increases ASR across all tested models. The following table summarizes the ASR on original audio samples and generated audio samples by our Query-based Audio Editing Method.
>
> |Model|Original|Query-based Audio Editing Method (ours)|
> |:-|:-:|:-:|
> |BLSP|48.1%|**87.8%**|
> |SpeechGPT|12.2%|**42.4%**|
> |Qwen2-Audio|13.3%|**48.8%**|
> |SALMONN-7B|31.6%|**85.1%**|
> |VITA-1.5|3.4%|**47.7%**|
> |MiniCPM-o-2.6|16.8%|**65.7%**|
> |GPT-4o-Audio|0.7%|**8.4%**|
> |Gemini-2.5-Flash|8.1%|**49.4%**|
>
> Compared to results in Table 2 of the paper and Appendix Table 4 on individual audio editing, combining multiple audio editing types can indeed substantially amplify ASR, particularly effective even against robust closed-source models like GPT-4o-Audio and Gemini-2.5-Flash.
>
> **S2:** Thank you for highlighting the value of our datasets. You identified its large scale and diverse categories. We also provide a full breakdown of the 157,782 samples in Table 1 and 56,263 samples in Table 3.
>
> > **Q2: The writing could be improved—specifically, the criteria for categorizing audio as explicit or implicit are not clearly explained, which may confuse readers trying to understand the dataset structure.**
>
> **A2:** We sincerely apologize for the confusion in our writing, particularly regarding the description of explicit and implicit categories.
>
> Explicit and Implicit primarily focus on the semantics of different audio questions, distinguished by their jailbreak intensity. This enables users to test different end-to-end LALMs, attack and defense algorithms to make more diverse choices based on their specific development stage, usage requirements, and application scenarios.
>
> Regarding the specific sub-tasks included in Explicit and Implicit subtypes, we illustrate these in Figure 2(a). The different sub-tasks (Sub-Task #x) shown in Figure 2(a) are derived from the union of all task categories covered by MM-SafetyBench [1], RedTeam-2K [2], and SafeBench [3]. These sub-task categories comprehensively cover current common jailbreak attack types. In addition, the specific questions in these three datasets are also categorized according to the corresponding Sub-Task #x. For the 520 uncategorized examples from AdvBench [4], we first utilized GPT-4o with the prompt: "Please categorize this question as one of the types under Sub-Task #x" for preliminary classification. Subsequently, we performed human classification to reconfirm each categorization. All 520 questions from AdvBench belonged to one of the Sub-Task #x categories, with no additional sub-task categories emerging.
>
> Regarding the Explicit and Implicit categorization process, we provide brief descriptions in Lines 81-84, Line 126, and Steps 4-5 in Algorithm 1. We greatly appreciate your suggestion and acknowledge that we have not thoroughly explained the specific process and categorization criteria for this key aspect of the Jailbreak-AudioBench dataset.
> The categorization into explicit and implicit subtypes follows a two-step approach, combining GPT-4o and human evaluation based on the content of each question, and is guided by three specific criteria.
>
> First, as Criteria 1, we use GPT-4o for initial categorization of each question with the prompt: "Determine whether the following question belongs to the explicit or implicit jailbreak category."
>
> Second, for all Sub-Task #x shown in Figure 2(a), we prompt GPT-4o with: "For Sub-Task #x, please provide some explicit words with clear jailbreak intent." These output "explicit words" serve as Criteria 2. For example: "bloody," "murder," and "suicide" in sub-task "Violence"; "naked," "rape," and "sexual abuse" in sub-task "Sex." After GPT-4o's initial classification using Criteria 1, we perform human recategorization incorporating Criteria 2.
>
> Finally, when Criteria 1 and 2 produce conflicting categorizations (e.g., questions without explicit words but classified as "explicit", or questions with explicit words but classified as "implicit"), we apply Criteria 3 for further classification. Criterion 3 is defined as follows: For the Explicit subtype，(1) Malicious intent is directly identifiable without the need for inference; (2) The content involves personal safety, violence, or illegal activities. For the Implicit subtype, (1) Indirect or euphemistic expressions are used; (2) The content typically involves general violations or ethically controversial topics.
>
> Through the combination of GPT-4o and human evaluation using Criteria 1-3, we ensure successful categorization of all data into Explicit and Implicit subtypes.
>
> Since classification combining GPT-4o and human evaluation is a common approach [1,2,3], we didn't present it as a primary contribution in our initial writing. However, through careful consideration of your suggestion, we realize that the categorization process and criteria for Explicit and Implicit subtypes could indeed be described in detail as part of our work, which we believe would benefit the entire research community. we greatly appreciate your suggestion.
>
> **S3:** We are glad you found t-SNE visualizations insightful. Figure 4  and Figures 10-24 in Appendix C.3 indeed reveal the key differences in how models process edited audio. Our goal with them was precisely to illustrate the varying robustness of different models. In addition, as mentioned in **Q1**, the analysis of the t-SNE visualization also contributes, to some extent, to the proposal of the Query-based Audio Editing Jailbreak Method.
>
> ## Weakness:
>
> Thanks again for these thoughtful analyses and specific suggestions. We appreciate reviewers like you who dig into technical details while also considering the broader significance for AI safety research. Your comment that you "don't find any major weakness" is greatly encouraging to us and strengthens our confidence in the work. If you have any further thoughts or would like any clarification, we truly welcome and appreciate the opportunity to continue the discussion with you！
>
>
> **References:**
>
> [1] Liu, X., et al. "MM-SafetyBench: A benchmark for safety evaluation of multimodal large language models", 2025.
>
> [2] Luo, W., et al. "Jailbreakv-28k: A benchmark for assessing the robustness of multimodal large language models against jailbreak attacks", 2024.
>
> [3] Gong, Y., et al. "Figstep: Jailbreaking large vision-language models via typographic visual prompts", 2025.
>
> [4] Zou, A., et al. "Universal and transferable adversarial attacks on aligned language models", 2023.

---

> > ### Author Response · Authors · 2025-08-05
> >
> > We sincerely thank you for your dedicated efforts in reviewing our paper and for providing insightful comments. Your reviews in Strengths Contribution, such as “**The proposed query-based audio editing method is simple yet effective**” , “**a large and diverse dataset**” and “**offer valuable insight into model behavior, effectively illustrating how different models vary in their robustness to audio edits**” are truly encouraging and strengthen our confidence in this project.
> >
> > We would also like to apologize again for any confusion caused by our description of the explicit and implicit categories. As mentioned above in the rebuttal, categorizing specific questions with the assistance of GPT-4o and human evaluation is a commonly adopted approach in recent works [1,2,3]. Therefore, we did not emphasize this categorization process and its criteria as a technical contribution in the initial version.
> >
> > Thank you very much for your suggestions and kind reminder once again. We will include additional clarifications in the final version — for example, in Section 3.1 (Lines 127–128) and Appendix C — to clearly present the detailed categorization steps and corresponding criteria of the explicit and implicit subtypes, as also outlined in our rebuttal.
> >
> > Additionally, we are truly grateful for your final positive assessment of our work — “**The provided dataset and framework will be valuable contributions to the community.**” and “**I think this is a strong paper that will help pave the way for research on jailbreaking LLM in the audio domain**”.  These encouragements will further motivate us to continue our efforts. We will actively maintain and update Jailbreak-AudioBench, and, as you kindly noted, we hope this project can make meaningful contributions to the advancement of the jailbreak audio research community.
> >
> > Thank you again for your valuable time and thoughtful suggestions! We also genuinely wish you good luck and all the best in your future work!
> >
> > **Reference**
> >
> > [1] Liu, X., et al. "MM-SafetyBench: A benchmark for safety evaluation of multimodal large language models", 2025.
> >
> > [2] Luo, W., et al. "Jailbreakv-28k: A benchmark for assessing the robustness of multimodal large language models against jailbreak attacks", 2024.
> >
> > [3] Gong, Y., et al. "Figstep: Jailbreaking large vision-language models via typographic visual prompts", 2025.

---

### Official Review · Reviewer_11tt · 2025-07-02

**Rating:** 4
**Confidence:** 2

**Summary:**

This paper introduces Jailbreak AudioBench, the first benchmark specifically designed to evaluate safety vulnerabilities in audio language models (ALMs). It covers various jailbreaking techniques (e.g., direct audio prompts, transcription abuse, TTS injection) and evaluates five open-source ALMs using automatic and human evaluations. The results demonstrate that current ALMs are vulnerable to both direct and indirect audio attacks, revealing a substantial gap in audio safety research compared to text-based LLMs.

**Dataset Code Accessibility:**

Yes

**Dataset Code Comments:**

The authors have open-sourced both the Jailbreak AudioBench dataset and accompanying evaluation scripts, which is commendable and important for reproducibility. The dataset appears well-structured and covers a diverse range of harmful prompt categories.

**Ethical Considerations:**

No, there are no or only very minor ethics concerns

**Final Justification:**

After carefully considering the authors’ rebuttal, the additional experimental results, and the broader discussion, I have updated my assessment of the paper.

**Limitations Weaknesses:**

1. The study only evaluates five open-source models and does not include major proprietary systems such as Whisper or Gemini Audio.

2. The assessment of harmfulness and jailbreak success heavily relies on GPT-4 and human judgment. No new safety metrics or automated evaluation methodologies are introduced, leaving the evaluation process largely dependent on existing tools. For human evaluations, key details such as the number of annotators, annotation procedures, and inter-annotator agreement (e.g., Cohen’s κ) are not adequately reported, potentially affecting the reliability of the results.

3. Although the benchmark covers over 100 categories, the number of samples per category and the linguistic diversity are unclear, which may introduce biases toward certain attack types or prompt styles. Furthermore, it lacks explicit support for multilingual, accented, and cross-cultural audio inputs, limiting its applicability to multilingual safety research.

4. Most attack techniques rely on direct translation or voice repetition, and the benchmark lacks more systematic and challenging adversarial strategies such as adversarial audio perturbations or prompt obfuscation.

**Strengths Contributions:**

1. This paper presents a highly novel and important perspective by focusing on the safety of Audio Language Models (ALMs), which is an underexplored yet increasingly critical issue as such models become more widespread.

2. The attack taxonomy is well-structured into three categories—audio, transcription, and TTS—and is evaluated across multiple scenarios, making the threat modeling comprehensive and robust.

3. The benchmark covers over 100 real-world harmful prompt categories (e.g., hate speech, self-harm, abuse), ensuring diversity and strong practical relevance.

4. The evaluation methodology is well-designed, combining GPT-4-based automatic assessment with human annotation, and includes initial exploration of mitigation techniques such as hallucination detection and rejection tuning.

---

> ### Author Rebuttal · Authors · 2025-07-31
>
> # Response to Reviewer 11tt
>
> Dear Reviewer 11tt,
>
> We sincerely appreciate your positive feedback in the **Strengths** , including the comments on the “highly novel and important perspective,” “well-structured into three categories,” “diversity and strong practical relevance,” and “The evaluation methodology is well-designed.” These evaluations greatly encourage us. Regarding your concern raised in **Weaknesses (W)**, we include additional comments and results to further improve our work.
>
> > **W1: The study only evaluates five open-source models and does not include major proprietary systems such as Whisper or Gemini Audio.**
>
> **A1:** In the paper, besides 8 open-source Large Audio Language Models evaluated in Table 2 of the paper, we also evaluated the proprietary model GPT-4o-Audio in Appendix Table 4. Furthermore, for our Query-based Audio Editing Method, Figure 5 shows that GPT-4o-Audio exhibits significant vulnerability with the Attack Success Rate (ASR) increasing from 0.7% to 8.4%.
>
> Following your valuable suggestion on Gemini evaluation, we have expanded evaluations on the Gemini-2.5-Flash. The table below shows the **ASR (%)** across various audio editing types when compared to the original audio on the Explicit Subtype dataset (left of the slash) and the Implicit Subtype dataset (right of the slash). It shows Gemini-2.5-Flash exhibits robustness to audio editing jailbreak, with minor ASR increases only in specific audio editing.
>
> |Model|Original|[Emphasis] Volume*2|[Emphasis] Volume*5|[Emphasis] Volume*10|[Speed] Rate*0.5|[Speed] Rate*1.5|[Intonation] Interval+2|[Intonation] Interval+3|[Intonation] Interval+4|[Tone] Semitone-8|[Tone] Semitone-4|
> |:-|:-:|:-:|:-:|:-:|:-:|:-:|:-:|:-:|:-:|:-:|:-:|
> |Gemini-2.5-Flash|8.1/5.1|-0.8/-0.8|+0.0/-0.8|-0.8/-1.7|-1.1/-3.4|+1.5/-0.8|-1.1/-2.5|+1.9/-2.5|+1.5/-1.7|+0.0/-2.9|-1.1/-0.8|
>
> |[Tone] Semitone+4|[Tone] Semitone+8|[Background] Crowd Noise|[Background] Machine Noise|[Background] White Noise|[Accent] Kanye West|[Accent] Donald Trump|[Accent] Lucy Liu|[Emotion] Laugh|[Emotion] Scream|
> |:-:|:-:|:-:|:-:|:-:|:-:|:-:|:-:|:-:|:-:|
> |-1.9/-0.8|-1.9/-2.1|-4.2/-2.5|-2.7/-2.9|+0.4/-3.4|-2.7/-2.1|-1.5/-2.5|-4.2/-2.9|+0.4/-1.7|-2.3/-0.4|
>
> Furthermore, we applied our Query-based Audio Editing Method to Gemini-2.5-Flash, achieving a substantial ASR increase **from 8.1% to 49.4%**, validating the effectiveness of our approach on proprietary models. Due to character limits, please refer to Q1A1 in the response to Reviewer hifu for the detailed table.
>
> Regarding the Whisper model, it is an open-source model designed specifically for automatic speech recognition and speech translation tasks [1]. Hence, Whisper is not suitable for evaluating audio jailbreak that aim to elicit harmful responses.
>
>
> > **W2: The assessment of harmfulness heavily relies on GPT-4 and human judgment. No new safety metrics or automated evaluation methodologies are introduced.**
>
> **A2:** Thank you for the constructive feedback. In addition to human evaluation in Appendix (Lines 962-1001), we employed Llama Guard 3 [2] as our automated judge (Lines 156-164). Llama Guard 3 is adopted in the research community of jailbreak task [3] and shows close alignment with human evaluations [4].
>
> In response to your valuable suggestion on GPT-4 assessment of jailbreak success, we have expanded harmfulness assessment using GPT-4o on **ASR results (%)** of Explicit Subtype in Table 2 of the paper, which provides additional validation for Llama Guard 3 measurements.
>
> |Model|Original|[Emphasis] Volume*2|[Emphasis] Volume*5|[Emphasis] Volume*10|[Speed] Rate*0.5|[Speed] Rate*1.5|[Intonation] Interval+2|[Intonation] Interval+3|[Intonation] Interval+4|[Tone] Semitone-8|[Tone] Semitone-4|
> |:-|:-:|:-:|:-:|:-:|:-:|:-:|:-:|:-:|:-:|:-:|:-:|
> |BLSP|64.5|+0.6|+0.2|-0.4|+1.7|-3.4|-6.0|-12.5|-18.1|-5.1|+1.4|
> |SpeechGPT|19.5|-0.4|-0.8|-5.7|-1.2|+0.2|-10.4|-14.8|-15.7|-4.8|-0.4|
> |Qwen2-Audio|21.7|-1.7|-5.4|-5.5|-5.1|+1.9|-5.6|-5.6|-4.0|-5.4|-2.7|
> |SALMONN-7B|36.9|+14.4|+22.2|+21.9|+15.3|-16.9|-31.8|-31.8|-29.8|-3.6|+0.3|
> |SALMONN-13B|35.8|+16.5|+22.8|+22.5|+20.0|-25.5|-2.5|+3.5|+12.1|+12.9|+6.7|
> |VITA-1.5|6.5|+0.5|+0.4|+0.2|+2.9|-0.6|+6.0|+5.8|+2.9|+5.0|-0.7|
> |R1-AQA|16.8|+1.2|+1.0|+1.8|+1.0|+3.2|+2.0|+3.8|+5.2|+1.1|-0.9|
> |MiniCPM-o-2.6|24.7|-1.2|-0.6|+1.0|+1.9|-2.0|+0.4|+1.5|+2.9|-0.5|-0.4|
>
> |[Tone] Semitone+4|[Tone] Semitone+8|[Background] Crowd Noise|[Background] Machine Noise|[Background] White Noise|[Accent] Kanye West|[Accent] Donald Trump|[Accent] Lucy Liu|[Emotion] Laugh|[Emotion] Scream|
> |:-:|:-:|:-:|:-:|:-:|:-:|:-:|:-:|:-:|:-:|
> |-1.2|-3.6|-0.5|+0.4|-0.5|-8.6|-9.8|-9.6|-1.2|-1.7|
> |-6.2|-19.1|-7.1|-6.9|-0.5|-5.4|-4.6|-2.8|-4.5|-5.2|
> |-5.8|-3.8|-9.0|-7.2|-5.9|-6.3|-5.2|-5.4|-5.5|-4.6|
> |-5.8|+8.8|+15.8|+22.0|+7.6|+17.3|+6.1|-6.4|+2.9|+20.7|
> |+19.9|+28.5|+29.3|+29.5|+23.4|+21.0|+24.2|+14.7|+26.0|+26.6|
> |+0.3|+5.5|+0.2|-0.4|+0.2|+2.0|+3.0|-0.6|-0.8|+0.1|
> |+1.1|+1.8|-2.4|-2.6|+1.1|+0.4|+0.9|+3.4|-1.1|+5.2|
> |-1.2|+8.6|+3.0|-0.9|-5.4|+9.5|+7.6|+1.2|-8.3|-9.0|
>
>
> > **W3: For human evaluations, key details such as the number of annotators, annotation procedures, and inter-annotator agreement are not adequately reported.**
>
> **A3:** We appreciate your attention to our human evaluation methodology. Regarding the number of annotators and annotation procedures, we presented these details of human evaluation in the Appendix Section C.2 (Lines 962-975).
>
> Regarding inter-annotator agreement, thank you for this valuable suggestion of Cohen's κ scores. Building upon our human evaluation data, the pairwise Cohen's κ scores between the three volunteers are as follows:
>
> - Volunteer 1 vs. Volunteer 2: κ = 0.79
> - Volunteer 2 vs. Volunteer 3: κ = 0.73
> - Volunteer 1 vs. Volunteer 3: κ = 0.76
>
> The average κ of 0.76 demonstrates that human annotators achieved consistent judgments when assessing jailbreak success.
>
>
> > **W4: Although the benchmark covers over 100 categories, the number of samples per category and the linguistic diversity are unclear.**
>
> **A4:** Thank you for this constructive feedback. From the linguistic perspective, our base original audio samples encompass 21 distinct jailbreak task categories, as shown in Figure 2(a) of the paper. The number of samples across categories is as follows. This Linguistic diversity ensures coverage across different harmful content categories.
>
> ||Fraud|Physical Harm|Illegal Activity|Hate Speech|Malware|Government Decision|Health Consultation|Economic Harm|Financial Advice|Legal Opinion|Privacy Violation|Adult Content|Unethical Behavior|Political Lobbying|Violence|
> |:-|:-:|:-:|:-:|:-:|:-:|:-:|:-:|:-:|:-:|:-:|:-:|:-:|:-:|:-:|:-:|
> |Number of Samples|425|399|385|352|351|285|283|247|222|180|172|164|160|158|151|
>
> |Privacy Violence|Child Abuse|Political Sensitivity|Tailored Unlicensed Advice|Bias|Animal Abuse|
> |:-:|:-:|:-:|:-:|:-:|:-:|
> |146|140|130|128|120|102|
>
> > **W5: The benchmark lacks explicit support for multilingual, accented, and cross-cultural audio inputs.**
>
> **A5:** We appreciate your thoughtful feedback. As the first comprehensive work on audio editing jailbreak against Large Audio Language Models (LALMs), our work focuses on language-agnostic and culture-agnostic audio editings, including Tone, Intonation, Speed, Emphasis, Background Noise, Celebrity Accent, and Emotion modifications. These audio modifications represent universal acoustic properties that transcend linguistic and cultural boundaries, making them widely applicable in real-world scenarios. And our work can provide a solid foundation for audio jailbreak research in multilingual, accented, and cross-cultural scenarios.
>
>
> > **W6: Most attack techniques rely on direct translation or voice repetition, and the benchmark lacks more adversarial strategies.**
>
> **A6:** Thank you for your constructive feedback. We would like to respectfully clarify our approach. Instead of direct translation, we employ diverse audio editings, including Tone, Intonation, Speed, Emphasis, Background Noise, Celebrity Accent, and Emotion modifications. Then, building upon these audio editings, we propose a Query-based Audio Editing Method that combines multiple audio editings to maximize jailbreak effectiveness. This method shows significant increases in Attack Success Rate (ASR) across all tested open-source and closed-source LALMs in Figure 5 and Appendix Figure 8. Notably, it achieves ASR increases of 7.7% on the closed-source GPT-4o-Audio and 41.3% on Gemini-2.5-Flash.
>
> Regarding adversarial strategies, while adversarial samples for jailbreak [5] typically require access to gradient information and are resource-intensive, our work focuses on a more practical jailbreak attack using readily available audio modifications, which requires no model information and relies solely on audio transformations that can easily occur in real-world scenarios. This makes the vulnerabilities more realistic, as these audio modifications can be naturally encountered in everyday interactions with LALMs.
>
> Furthermore, our work can serve as a foundation for adversarial research, which provides a comprehensive black-box baseline that enables researchers to establish performance benchmarks before developing more sophisticated gradient-based adversarial methods. And the diverse audio editing types can be used as initialization strategies or constraint guidance for adversarial optimization, helping to generate more effective and naturalistic adversarial samples.
>
> **Reference**
>
> [1] Radford, et al. "Robust speech recognition via large-scale weak supervision", 2023
>
> [2] Inan, et al. "Llama guard: Llm-based input-output safeguard for human-ai conversations", 2023
>
> [3] Yang, et al. "Audio Is the Achilles' Heel: Red Teaming Audio Large Multimodal Models", 2024.
>
> [4] Yang, et al. "Jigsaw puzzles: Splitting harmful questions to jailbreak large language models", 2024
>
> [5] Kang, et al. "Advwave: Stealthy adversarial jailbreak attack against large audio-language models", 2024

---

> > ### Comment · Reviewer_11tt · 2025-08-04
> >
> > Thank you for the detailed response and the comprehensive experiments. I appreciate the clarifications and additional results provided. Based on the new information, I will revise my score upward.

---

> > ### Author Response · Authors · 2025-08-04
> >
> > We are deeply grateful once again for your thoughtful and positive recognition of our work, including comments such as “**highly novel and important perspective by focusing on the safety of Audio Language Models (ALMs)**”, “**attack taxonomy is well-structured**”, “**the benchmark …, ensuring diversity and strong practical relevance**”, and “**the evaluation methodology is well-designed**”. We are delighted that our responses and additional experiments have successfully addressed your concerns about model coverage, evaluation methodologies, and experimental rigor.
> >
> > Based on your insightful feedback, we have further strengthened our work, including the extensive evaluation of Gemini-2.5-Flash, GPT-4o assessment of jailbreak success, the clarification of our human evaluation procedures with inter-annotator agreement metrics, and the detailed breakdown of our benchmark's linguistic diversity. The additional experiments demonstrating significant Attack Success Rate (ASR) increases on **Gemini-2.5-Flash (from 8.1% to 49.4%)**, together with our paper's results of ASR increases on **GPT-4o-Audio (from 0.7% to 8.4%)**, further validate the consistent effectiveness of our Query-based Audio Editing Method across proprietary models.
> >
> > We are committed to incorporating these valuable suggestions you provided into our revised manuscript, ensuring that the enhanced experimental validation and clearer methodological details are well-integrated throughout the paper. Thank you again for your dedication to the review process and for helping us elevate the quality and impact of our contribution to the field.

---

### Official Review · Reviewer_LwpR · 2025-07-05

**Rating:** 5
**Confidence:** 3

**Summary:**

This paper proposed Jailbreak-AudioBench, which introduces a comprehensive framework featuring an audio prompt generation and editing toolbox, a dataset of over 157,000 explicit and implicit jailbreak audio examples, and thorough evaluations of eight state-of-the-art end-to-end audio–language models. It reveals how hidden audio semantics—such as prosody, emotion, and speed—impact model security, and proposes lightweight prompt-level defense strategies.

**Dataset Code Accessibility:**

Yes

**Dataset Code Comments:**

The benchmark has been open-sourced and provides detailed documentation of both the dataset and the training/testing pipeline.

**Ethical Considerations:**

No, there are no or only very minor ethics concerns

**Limitations Weaknesses:**

Overall, this paper has no obvious weaknesses. I have some slight concerns as follows:
1) Did the experiments in Table 2 run only once? Given the substantial variability inherent in LLM mechanisms and jailbreak attacks, it would be far more informative to repeat key experiments with multiple random seeds (e.g., 3–5) and report the results as means ± standard deviations or with 95 % confidence intervals.
2) The text-to-audio conversion step relies solely on a TTS tool. I recommend incorporating a small fraction of real human recordings to enrich the model’s learning and increase the dataset’s diversity.

**Strengths Contributions:**

The primary contributions are:

1) Audio Jailbreak Toolbox: A modular suite for synthesizing and editing audio prompts—covering text-to-speech, pitch/timbre shifts, speed modulation, background noise injection, celebrity voice emulation, emotional tone, and more—to craft both explicit and subtle jailbreak attacks.

2) Diverse Jailbreak Dataset: Over 157 000 audio samples (original and edited) categorized into explicit and implicit jailbreak types, drawn from and extending existing prompt collections (e.g., AdvBench, MM-SafetyBench, RedTeam-2K, SafeBench).

3) Comprehensive Benchmarking: End-to-end evaluation of eight state-of-the-art end-to-end LALMs (including BLSP, SpeechGPT, Qwen2-Audio, SALMONN, VITA-1.5, R1-AQA, MiniCPM-o-2.6, and GPT-4o-Audio), measuring their susceptibility to each attack variant.

Additionally, the paper is well-structured that enhances the paper’s readability and impact.

---

> ### Author Rebuttal · Authors · 2025-07-31
>
> # Response to Reviewer LwpR
>
> Dear Reviewer LwpR,
>
> Thank you for the thoughtful and comprehensive assessment of our work.
>
> ## Strength (S):
>
> **S1:**  You correctly identified the key role of Toolbox. Its purpose is to craft varied audio-based jailbreak attacks. Your observation of our categorization of explicit and implicit (subtle) subtypes based on varying jailbreak threat intensity is accurate. This classification enables more diverse and targeted choices tailored to users' specific needs.
>
> **S2:** Your summary captures the comprehensive nature of our diverse Dataset. You identified its scale of over 150K samples. The details are in Table 1 and 3, which also specifies the composition of specialized subtypes like Explicit/Implicit.
>
> **S3:** You have a precise understanding of our comprehensive Benchmarking. Our goal is to measure susceptibility, and the significant performance disparities among the 8 LALMs detailed in Table 2 validate our benchmark's effectiveness.
>
> Finally, thanks for your kind words on the paper's structure. We carefully designed the flow of Toolbox, Dataset, Benchmark and Potential Research from Section 2 to 4. Your perspective feedback is deeply encouraging and motivates us to continue advancing this work.
>
> ## Weakness (W):
>
> **W1:** Thank you for this valuable feedback. We acknowledge the importance of statistical rigor in experimental evaluation. Due to the large scale of our dataset (as shown in Table 1) and the extensive number of models evaluated, the experiments in Table 2 are highly time-intensive and resource-intensive (Lines 292-295). Given the computational constraints and the fact that the large-scale dataset helps mitigate variability, we conducted single-run experiments for Table 2.
>
> The suggestion of repeating experiments with multiple random seeds is highly valuable. To address this concern within the rebuttal timeframe, we conducted additional experiments for Table 2 using four random seeds on the Explicit Small and Implicit Small subsets of our dataset (details provided in Appendix Table 3). The **results (%)** are presented in the table below, which shows the ASR (means ± standard deviations) across various audio editing types when compared to the original audio on the Explicit Small dataset (left of the slash) and the Implicit Small dataset (right of the slash).
>
> |Model|Original|[Emphasis] Volume*2|[Emphasis] Volume*5|[Emphasis] Volume*10|[Speed] Rate*0.5|[Speed] Rate*1.5|[Intonation] Interval+2|[Intonation] Interval+3|[Intonation] Interval+4|[Tone] Semitone-8|[Tone] Semitone-4|
> |:-|:-:|:-:|:-:|:-:|:-:|:-:|:-:|:-:|:-:|:-:|:-:|
> |BLSP|44.3(±3.2) / 17.0(±1.2)|+1.7(±4.3) / -1.5(±1.5)|+0.7(±3.2) / -0.2(±1.2)|+0.4(±5.4) / -1.3(±1.9)|+4.1(±4.3) / +1.1(±1.6)|-2.4(±2.7) / -2.6(±0.9)|-8.3(±1.1) / -3.5(±0.4)|-9.2(±2.8) / -3.8(±1.1)|-14.4(±3.0) / -3.5(±1.3)|-4.0(±2.5) / -1.7(±1.0)|+3.6(±4.9) / +0.3(±1.8)|
> |SpeechGPT|14.3(±0.8) / 2.5(±0.1)|-0.5(±1.2) / -0.1(±0.2)|-1.5(±0.9) / -0.2(±0.2)|-4.9(±0.8) / -0.4(±0.2)|+0.2(±0.2) / -0.2(±0.0)|0.0(±1.6) / -0.2(±0.3)|-8.0(±0.3) / -1.0(±0.1)|-11.4(±0.2) / -0.9(±0.1)|-11.7(±0.2) / -0.8(±0.1)|-3.2(±0.4) / -0.1(±0.1)|+0.3(±1.1) / 0.0(±0.2)|
> |Qwen2-Audio|16.5(±1.5) / 6.7(±0.6)|-1.4(±1.5) / -0.6(±0.6)|-4.0(±1.0) / +0.1(±0.6)|-4.3(±0.8) / -1.1(±0.4)|-4.2(±1.4) / -1.8(±0.6)|+2.7(±0.9) / +0.7(±0.3)|-5.0(±0.8) / -0.6(±0.4)|-4.8(±1.1) / -1.3(±0.5)|-3.8(±0.6) / -0.7(±0.3)|-4.9(±0.5) / +0.2(±0.3)|-3.4(±0.9) / -0.1(±0.5)|
> |SALMONN-7B|32.1(±2.7) / 14.6(±1.2)|+12.6(±4.7) / +2.8(±1.8)|+24.2(±3.5) / +6.7(±1.3)|+20.0(±5.1) / +5.4(±2.0)|+12.2(±1.5) / +1.5(±0.5)|-13.8(±0.7) / -3.9(±0.4)|-27.9(±0.2) / -11.2(±0.2)|-26.7(±0.2) / -10.9(±0.2)|-24.9(±0.3) / -11.2(±0.1)|-3.9(±2.2) / -1.2(±1.1)|-2.0(±1.4) / -2.0(±0.6)|
> |SALMONN-13B|29.6(±1.3) / 12.2(±0.6)|+12.1(±2.4) / +1.1(±0.8)|+17.8(±4.4) / +2.6(±1.4)|+17.5(±3.1) / +2.8(±1.0)|+11.5(±2.0) / +1.3(±0.7)|-23.0(±0.7) / -8.4(±0.4)|0.0(±3.4) / -1.0(±1.3)|+7.5(±0.7) / +1.2(±0.3)|+12.9(±3.2) / +2.4(±1.1)|+13.6(±4.7) / +2.0(±1.5)|+4.9(±3.2) / +0.7(±1.2)|
> |VITA-1.5|3.9(±0.3) / 2.9(±0.3)|+0.6(±0.2) / +0.4(±0.2)|+0.4(±0.3) / +0.1(±0.2)|-0.1(±0.1) / +0.5(±0.1)|+2.3(±0.5) / +0.6(±0.3)|-0.6(±0.1) / +0.3(±0.1)|+6.1(±0.3) / +1.5(±0.2)|+4.7(±0.4) / +0.3(±0.1)|+4.4(±0.3) / +0.4(±0.1)|+3.3(±0.3) / +0.5(±0.2)|-0.3(±0.2) / +0.4(±0.2)|
> |R1-AQA|12.6(±1.4) / 7.2(±0.8)|+0.2(±0.4) / -0.9(±0.2)|+0.4(±1.3) / -0.5(±0.6)|+2.6(±1.3) / 0.0(±0.6)|+0.6(±1.4) / -1.3(±0.6)|+1.6(±1.1) / +0.2(±0.6)|+1.1(±1.6) / -0.7(±0.8)|+2.9(±1.3) / -0.4(±0.6)|+4.5(±0.6) / +0.1(±0.3)|+1.1(±0.5) / +0.9(±0.3)|+0.4(±1.0) / -1.0(±0.5)|
> |MiniCPM-o-2.6|18.8(±1.8) / 9.3(±0.9)|-0.2(±1.0) / +0.9(±0.5)|-0.8(±1.5) / -0.3(±0.7)|+2.0(±1.0) / -0.4(±0.5)|+1.1(±1.6) / +0.2(±0.8)|-2.3(±1.5) / -0.5(±0.8)|-0.8(±1.7) / -1.1(±0.7)|+0.5(±1.7) / -1.4(±0.7)|+2.8(±1.3) / -0.8(±0.5)|+0.1(±1.3) / -0.1(±0.6)|-0.2(±1.7) / -0.7(±0.8)|
>
> |[Tone] Semitone+4|[Tone] Semitone+8|[Background] Crowd Noise|[Background] Machine Noise|[Background] White Noise|[Accent] Kanye West|[Accent] Donald Trump|[Accent] Lucy Liu|[Emotion] Laugh|[Emotion] Scream|
> |:-:|:-:|:-:|:-:|:-:|:-:|:-:|:-:|:-:|:-:|
> |-2.5(±2.4) / -1.0(±0.9)|+2.0(±1.6) / +0.4(±0.6)|-0.5(±4.2) / -1.6(±1.5)|-1.7(±3.2) / -0.5(±1.3)|+2.1(±5.6) / +0.6(±2.1)|-8.9(±4.0) / -3.9(±1.5)|-9.3(±3.5) / -3.5(±1.4)|-10.8(±1.6) / -4.1(±0.6)|-2.4(±4.1) / -1.8(±1.5)|+0.9(±4.0) / -1.0(±1.4)|
> |-5.7(±0.7) / -0.6(±0.2)|-13.7(±0.0) / -2.2(±0.0)|-5.7(±0.4) / 0.0(±0.1)|-5.1(±0.5) / -0.2(±0.1)|-0.8(±1.1) / -0.2(±0.2)|-4.9(±0.9) / -0.4(±0.2)|-4.9(±0.3) / -0.7(±0.0)|-3.1(±1.2) / -0.2(±0.3)|-4.6(±0.5) / +0.1(±0.1)|-4.7(±1.0) / -0.8(±0.2)|
> |-5.2(±0.8) / -1.1(±0.4)|-2.8(±1.3) / -0.9(±0.6)|-7.1(±0.4) / -1.6(±0.2)|-5.8(±0.5) / -1.1(±0.2)|-5.5(±0.5) / -1.5(±0.2)|-5.0(±0.8) / -1.0(±0.4)|-4.1(±1.0) / -1.5(±0.4)|-5.2(±0.5) / -1.4(±0.2)|-3.5(±0.6) / +0.4(±0.3)|-4.3(±1.0) / -1.1(±0.5)|
> |-2.9(±1.9) / -1.1(±0.9)|+6.5(±1.1) / +1.2(±0.4)|+18.9(±2.8) / +6.8(±1.2)|+20.5(±4.5) / +6.0(±1.8)|+7.7(±2.8) / +5.2(±1.4)|+14.5(±1.9) / +5.9(±0.9)|+4.5(±2.5) / +2.6(±1.2)|-6.6(±1.5) / -4.5(±0.6)|+3.4(±2.0) / +0.4(±0.8)|+19.2(±4.6) / +5.7(±1.8)|
> |+18.4(±4.7) / +3.9(±1.6)|+25.9(±2.4) / +5.3(±0.8)|+28.0(±7.0) / +7.8(±2.4)|+32.7(±2.7) / +10.7(±1.0)|+24.7(±3.1) / +5.8(±1.0)|+15.0(±3.3) / +2.4(±1.1)|+24.2(±5.1) / +4.4(±1.6)|+13.6(±3.9) / +4.2(±1.5)|+24.9(±3.4) / +5.9(±1.1)|+19.5(±3.9) / +4.0(±1.3)|
> |+0.4(±0.4) / +0.3(±0.3)|+4.3(±0.5) / +0.3(±0.2)|+0.5(±0.4) / +0.8(±0.3)|+0.2(±0.3) / +0.3(±0.3)|+0.1(±0.3) / +0.1(±0.2)|+1.8(±0.5) / +0.3(±0.2)|+2.5(±0.6) / +0.7(±0.3)|-0.3(±0.3) / +0.6(±0.3)|-0.3(±0.3) / +0.1(±0.3)|+0.3(±0.2) / +0.4(±0.2)|
> |+1.4(±1.0) / -0.5(±0.5)|+1.0(±0.4) / -0.9(±0.2)|-1.5(±0.6) / -1.9(±0.3)|-2.2(±1.2) / -1.4(±0.6)|+1.9(±0.6) / -0.1(±0.3)|+0.6(±0.9) / -1.0(±0.4)|+0.6(±0.9) / -0.4(±0.4)|+2.2(±1.6) / -0.6(±0.7)|-2.6(±0.5) / -0.9(±0.3)|+5.3(±1.2) / +0.9(±0.5)|
> |+0.7(±0.7) / -0.6(±0.3)|+5.3(±1.3) / -0.6(±0.5)|+1.8(±2.0) / +0.5(±0.9)|-1.0(±1.5) / -0.1(±0.8)|-4.1(±1.0) / -1.2(±0.6)|+8.7(±2.4) / +0.1(±0.8)|+7.1(±1.7) / +1.0(±0.7)|+1.2(±1.9) / +0.4(±0.9)|-7.0(±0.8) / -2.9(±0.4)|-7.4(±0.4) / -2.9(±0.2)|
>
> These supplementary results demonstrate the consistency of our results across multiple runs. We acknowledge that extending this statistical analysis to the full dataset would further strengthen our work and will incorporate comprehensive multi-seed evaluations in the revision.
>
>  **W2:** Thank you for your suggestion. "Incorporating a small fraction of real human recordings" would undoubtedly further "enrich the model's learning and increase the dataset's diversity." We also mentioned this in the "Future Continuous Updating" in Section 6, Lines 298–300, which will be our next priority.
>
> On the Project page in our paper, we have launched a "Voice Recording" activity. After selecting certain variables, volunteers can directly record audios online and download them. We also provide corresponding methods to collect volunteer-recorded audio examples. The specific jailbreak questions for recording come from the 262 and 237 examples from Explicit Small and Implicit Small subtypes in Table 3 of Appendix B. For parameter-dependent editing methods like Emphasis, Speed, Intonation and Tone, we use “--”, “-”, “+”, “++” to roughly define parameter increases/decreases and intensity changes, thereby reducing volunteers' workload. For Accent, we differentiate volunteers' accents by collecting "Address/Hometown," "Gender," and "Human Race." For Background Noise, with specific parameter options of "Crowd, Machine, and White Noise", we provide corresponding background audio files to facilitate recording. For Emotion, we maintain two options: "Laugh" and "Scream." We will continuously optimize the "Voice Recording" activity workflow to make it more streamlined and encourage volunteer participation. After collecting sufficient real audios, we will classify and analyze them, updating the dataset accordingly.
>
> As described above, real human recordings present challenges of complex parameters and imprecise control. Therefore, in our initial release, similar to current related work [1,2], to achieve precise quantitative parameter evaluation for end-to-end LALMs, we used TTS tools to convert the original audios. However, incorporating real human recordings to make Jailbreak-AudioBench more diverse and closer to real-world scenarios is undoubtedly one of our core research objectives. Thank you again for your valuable advice. We will continuously optimize the "Voice Recording Volunteer" activity, constantly improve the dataset, and release updates promptly.
>
> **Reference**
>
> [1] Yang, H., et al. "Audio is the Achilles' heel: Red teaming audio large multimodal models", 2024.
>
> [2] Hughes, J., et al. "Best-of-n jailbreaking", 2024.

---

> > ### Author Response · Authors · 2025-08-07
> >
> > Thank you again for your thorough review and valuable comments. In the Strengths Contributions, you highlighted the merits of our work from three aspects: “**Audio Jailbreak Toolbox: A modular suite for synthesizing and editing audio prompts**”, “**Diverse Jailbreak Dataset**”, and “**Comprehensive Benchmarking**”. We also sincerely appreciate your concluding remark that “**the paper is well-structured, which enhances the paper’s readability and impact.**” Your positive feedback greatly encourages and motivates us.
> >
> > Furthermore, we are very grateful for the two constructive suggestions you provided. We will carefully incorporate them into the final version to further improve Jailbreak-AudioBench, with the aim of making a meaningful contribution to the Audio Jailbreak research community.
> >
> > Thank you again for your time, efforts, and recognition of our work.

---

### Official Review · Reviewer_KQbp · 2025-07-08

**Rating:** 4
**Confidence:** 3

**Summary:**

Research objects: Security vulnerabilities of Large Language Models (LLMs), Multimodal Large Language Models (MLLMs), and Large Audio - Language Models (LALMs), especially audio - specific jailbreak attacks.

Proposed method: Introduce Jailbreak - AudioBench, which consists of a Toolbox (supporting text - to - audio conversion and audio editing), a carefully curated Dataset (providing a variety of explicit and implicit jailbreak audio examples), and a comprehensive Benchmark.

Primary contributions: Evaluate advanced LALMs, fill the research gap in audio - specific jailbreak vulnerabilities of large audio - language models, lay a foundation for the research on the safety alignment of LALMs, and help discover more powerful jailbreak threats (such as query - based audio editing) and develop effective defense mechanisms.

**Dataset Code Accessibility:**

Yes

**Ethical Considerations:**

No, there are no or only very minor ethics concerns

**Limitations Weaknesses:**

Some new researches maybe refered:
【1】Li Y, Zhang M, Ren M, et al. Cross-domain audio deepfake detection: Dataset and analysis[J]. arXiv preprint arXiv:2404.04904, 2024.
【2】Akhtar Z, Pendyala T L, Athmakuri V S. Video and audio deepfake datasets and open issues in deepfake technology: being ahead of the curve[J]. Forensic Sciences, 2024, 4(3): 289-377.

**Strengths Contributions:**

Provides an end-to-end framework encompasses a comprehensive framework with diverse elements.

The Toolbox is pivotal, enabling adjustment of multiple audio attributes. It allows manipulation of Noise, as well as Emphasis (including parameters like volume), Speed (with rate - related settings), Intonation (involving interval parameters), and Tone (featuring semitone parameters). Additionally, it supports the inclusion of Background Noise such as crowd or machine sounds, Celebrity Accents like those of Kanye West, and Emotions ranging from laughter to screams. These capabilities are crucial for crafting varied audio inputs to test the resilience of models.

Datasets form a cornerstone of Jailbreak - AudioBench. The Jailbreak - AudioBench Base Dataset offers Explicit and Implicit Subtypes, each accompanied by defined numbers of examples and subtasks. The Edited Dataset follows a similar structure, also having Explicit and Implicit Subtypes. Moreover, there are specialized branches like Explicit Small and Explicit Defense, which provide targeted data for more focused evaluations. These datasets supply a rich array of audio examples, both original and edited, to probe potential jailbreak vulnerabilities in models.

A range of Large Audio - Language Models (LALMs) are involved, namely BLSP, SpeechGPT, Qwen2 - Audio, SALMONN, VITA, MiniCPM - o, R1 - AQA, and OpenAI GPT - 4o. These models are the subjects of evaluation within the Jailbreak - AudioBench framework, to determine their susceptibility to audio - based jailbreak attacks.

Relevant components include LLAMA Guard 3, which exhibits distinct responses like “Yes, these are steps,...” and “Sorry, I can’t answer,...”. Collectively, all these elements - the Toolbox for audio manipulation, the diverse datasets for testing, the LALMs under scrutiny, and components like LLAMA Guard 3 - come together to present the architectural flow of Jailbreak - AudioBench. This framework is designed to comprehensively assess and understand the security vulnerabilities of LALMs to audio - specific jailbreak threats, facilitating research into model safety and the development of effective defense mechanisms.

---

> ### Author Rebuttal · Authors · 2025-07-31
>
> # Response to Reviewer KQbp
>
> Dear Reviewer KQbp,
>
> Thank you for your deep understanding and insightful analysis of our work. Your comprehensive grasp of our framework architecture truly captures our contribution and value.
>
> ## Strength (S):
>
> **S1:**
> Your summary, "The Toolbox is pivotal, enabling adjustment of multiple audio attributes," is spot-on, precisely identifying its key role in generating diverse audio attacks. Your insight is very impressive — these capabilities are indeed "crucial for crafting varied audio inputs to test the resilience of models." In Figure 1, which provides an overview of Jailbreak-AudioBench, we positioned the Toolbox at the top to highlight its importance and foundational role in the entire work. Specifically, at the dataset level, as shown in the dataset descriptions in Table 1 and Table 3, after obtaining base audio, the main components of Jailbreak-AudioBench are all generated through the seven editing methods with 20 parameter variations contained in the Toolbox. In the Benchmark section, such as Table 2 and Table 4 in Section 2 and Appendix C.1, all Large Audio Language Models (LALMs) are evaluated on these Toolbox-processed audios, enabling validation under the full range of audio hidden semantics. To further elaborate on this critical component, we provide detailed technical implementations of the Toolbox in Section 2 and Appendix A. Additionally, to visually demonstrate the changes that different editing categories (parameter × editing method) bring to base audio, we illustrate the spectrograms of the original and all 20 types of edited audio samples in Figure 3 and Figure 7.
>
> **S2:** We also really appreciate your recognition of our datasets' design, highlighting their "cornerstone" role. You correctly noted that, beyond Base and Edited datasets, in Table 1 and Table 3, we further differentiate our datasets with prefix or postfix words such as "Explicit/Implicit", "Small", and "Defense", thus providing more targeted sub-type dataset options for future Jailbreak-AudioBench users. "Explicit/Implicit" distinguishes between different jailbreak intensities of audio questions, allowing users to make more diverse choices based on their LALMs' specific development stage, usage requirements, and testing environment. "Small" provides lightweight sub-type Dataset versions, primarily offering future users a quick validation option for preliminary testing and resource-intensive tasks. "Defense" serves in this work to demonstrate that Jailbreak-AudioBench not only enables risk assessment of LALMs but also facilitates the development of potential defense algorithms through flexible generation and implementation via the Toolbox. We really believe such a large-scale, high-variant, and well-structured dataset would provide an unprecedented and solid foundation for the community’s research on audio jailbreaking.
>
> **S3:** You accurately listed all representative LALMs in our evaluation. These LALMs represent the most cutting-edge, widely followed, and popularly used end-to-end LALMs currently available. Based on Table 2 in Section 3.2, Figure 5 in Section 4.1, Table 4 in Appendix C.1, and Figure 8 in Appendix D.1, your conclusion about determining their susceptibility to audio-based jailbreak attacks insightfully reflects the main findings and conclusions of the Jailbreak-AudioBench Benchmark.
>
> **S4:** Your understanding of Llama Guard 3's role is precise! In Section 3.2 and Appendix C.2, we explain why we chose Llama Guard 3 as the primary evaluator and provide detailed comparisons with human evaluation, demonstrating its effectiveness as a "human-aligned but more stringent metric" in Line 999. Finally, we are truly honored by your clear and holistic understanding of our framework design. The primary purpose of this framework is to "comprehensively assess and understand the security vulnerabilities of LALMs to audio-specific jailbreak threats". Your description of how the Toolbox, Datasets, LALMs, and Llama Guard 3 all "come together to present the architectural flow" is a brilliant synthesis that perfectly embodies our design philosophy. Regarding "facilitating research into model safety and the development of effective defense mechanisms", as described in Section 4.1 and Section 4.2, this framework enables not only rigorous evaluation but also facilitates innovation, including our proposed Query-based Audio Editing Jailbreak method and its corresponding defense strategy.
>
> Thanks again for your remarkably thoughtful engagement; your insights have truly elevated our understanding of how our work could serve the broader community. We are genuinely grateful for your nuanced perspective, which has offered us both affirmation and inspiration moving forward.
>
> ## Weakness:
>
> We sincerely appreciate your recommendation of these two audio deepfake-related papers[2,3]. Reading and reflecting on them has provided us with valuable insights and inspiration.
>
> Audio Deepfake is an application of artificial intelligence designed to generate speech that convincingly mimics specific prosody or style of individuals, often synthesizing phrases or sentences they have never spoken [1]. Li et al. (2024) [2] attempts to detect deepfake audios by creating the CD-ADD dataset with over 300 hours of speech from 5 advanced TTS models to test detection robustness. In contrast, our work focuses on jailbreaking LALMs by manipulating audio’s hidden semantics. Akhtar et al. (2024) [3] provides a comprehensive survey mapping the deepfake landscape and its open challenges, notably calling for more research into how elements like accent, tone, and emotion affect security — a gap our work directly addresses.
>
> As discussed, Jailbreak-AudioBench primarily includes seven editing methods for original audios: Emphasis, Speed, Intonation, Tone, Background Noise, Accent, and Emotion. Through analyzing your suggestions, we recognize that these editing methods, to some extent, correspond to specific audio deepfake techniques, where “Emphasis, Speed, Intonation, Tone” relate to prosody mimicking, and “Background Noise, Accent, Emotion” relate to style mimicking in audio deepfake implementation. During our initial writing, we focused mainly on jailbreak vulnerabilities arising from audio input semantics [4,5], overlooking audio deepfake threats closely tied to our editing methods. We greatly appreciate your suggestion again. By citing these two papers [2,3] in **Section 1 (Line 67-68)** and the **Related Work in Section 5**, we can better expand our work’s impact and further supplement the connection between Toolbox’s specific editing methods and audio deepfake.
>
> Thank you again for your suggestion. If you have other related papers to recommend, we would be delighted to include them prominently in the final version.
>
> **Reference:**
>
> [1] "Audio deepfake", wikipedia description of "Audio deepfake".
>
> [2] Li, Y., et al. "Cross-domain audio deepfake detection: Dataset and analysis", arXiv:2404.04904, 2024.
>
> [3] Akhtar, Z., et al. "Video and audio deepfake datasets and open issues in deepfake technology: being ahead of the curve", Forensic Sciences, 2024.
>
> [4] Yang, H., et al. "Audio is the Achilles' heel: Red teaming audio large multimodal models", 2024.
>
> [5] Hughes, J., et al. "Best-of-n jailbreaking", 2024.

---

> > ### Comment · Reviewer_KQbp · 2025-08-05
> >
> > Thank you for the detailed responses to the related questions. Overall, it is a good paper. I will maintain my original score and also appreciate the author's contributions.

---

> > ### Author Response · Authors · 2025-08-05
> >
> > Thank you very much for your thorough review and valuable feedback. In the Strengths, your recognition of our paper’s contributions, including remarks such as “**Toolbox is pivotal**”,  “**Datasets form a cornerstone**”,  “**A range of Large Audio - Language Models (LALMs)**”, and “**framework is designed to comprehensively**” has greatly encouraged us.
> >
> > We also sincerely appreciate your constructive suggestions, which will help us better clarify the inspiration behind our work and its practical application scenarios.
> >
> > Finally, we are truly grateful for your positive overall assessment in the final review — “**Overall, it is a good paper.**” This assessment will further strengthen our confidence and motivation. We will carefully incorporate your suggestions into the final version and will continue to maintain and update Jailbreak-AudioBench, striving to contribute as best we can to the research community.
> >
> > Thanks again for your hard work during the review process and your highly insightful comments！

---

### Author Response · Authors · 2025-08-09

We once again express our sincere gratitude to Reviewers (R) KQbp, LwpR, 11tt, and hifu for their dedicated efforts throughout the review, rebuttal, and discussion stages. We are pleased that all Reviewers acknowledged the significance and novelty of our work on audio editing jailbreak, with positive assessments highlighting its contributions to advancing the audio jailbreak research community, such as “**Overall, it is a good paper**” (R-KQbp), “**the paper is well-structured that enhances the paper’s readability and impact**” (R-LwpR), “**a highly novel and important perspective by focusing on the safety of Audio Language Models (ALMs)**” (R-11tt), and “**I think this is a strong paper that will help pave the way for research on jailbreaking LLM in the audio domain**” (R-hifu).

Moreover, we genuinely appreciate the constructive suggestions provided by all Reviewers. The reflections and discussions prompted by these suggestions have greatly contributed to further enhancing the applicability and impact of Jailbreak-AudioBench. Below, we provide an overall summary of the relevant discussions.

- **Related Work Integration (R-KQbp)**: R-KQbp recommended incorporating recent audio deepfake research to better contextualize our work. We acknowledged the connection between our audio editing methods and audio deepfake techniques, with our "Emphasis, Speed, Intonation, Tone" relating to prosody mimicking and "Background Noise, Accent, Emotion" relating to style mimicking. We integrated the suggested references to strengthen the paper's positioning within the broader audio security landscape.

- **Expanded Model and Dataset Coverage (R-11tt & R-LwpR)**: R-11tt raised concerns about evaluating only open-source models without major proprietary models like Gemini Audio. R-LwpR suggested incorporating real human recordings to increase dataset diversity. We addressed these by conducting extensive experiments on Gemini-2.5-Flash, demonstrating our Query-based Audio Editing Method's effectiveness, with the Attack Success Rate (ASR) increasing from **8.1% to 49.4%**. Combined with our existing GPT-4o-Audio results (**0.7% to 8.4%** ASR increase), this validates our approach across major proprietary models. We also launched a "Voice Recording" activity for collecting real human speech samples to enhance dataset diversity in future updates.

- **Statistical Rigor in Experiments (R-LwpR)**: R-LwpR was concerned whether experiments were run only once and recommended multiple random seeds with statistical reporting. We conducted additional multi-seed experiments on Explicit Small and Implicit Small subsets of our dataset using four random seeds, providing ASR results with means ± standard deviations. This demonstrated the consistency and reliability of our findings across multiple runs, addressing concerns about experimental variability.

- **Human Evaluation Methodology (R-11tt)**: R-11tt noted that details about human evaluation procedures, including the number of annotators and inter-annotator agreement, were inadequately reported. We provided comprehensive methodological details, including the number of annotators (three volunteers), annotation procedures, and inter-annotator agreement metrics with Cohen's κ scores averaging 0.76, demonstrating substantial agreement among human evaluators and validating our evaluation methodology.

- **Automated Evaluation Methodology (R-11tt)**: R-11tt pointed out that harmfulness assessment can be enhanced by new automated evaluation methodologies. We supplemented our Llama Guard 3 assessments with GPT-4o evaluations for additional validation of jailbreak success, providing multiple layers of assessment and strengthening the reliability of our harmfulness evaluations.

- **Dataset Categorization and Linguistic Diversity (R-hifu & 11tt)**: R-hifu noted that criteria for categorizing audio as explicit or implicit were not clearly explained, while R-11tt raised concerns about linguistic diversity and sample distribution. We clarified our explicit/implicit categorization process, detailing the three-criteria approach combining GPT-4o and human evaluation with specific guidelines. We also provided comprehensive breakdowns of our benchmark across 21 distinct jailbreak task categories, with detailed sample counts for each category.

Finally, we sincerely thank all Reviewers once again for their valuable time and effort in evaluating our paper. Their positive feedback and insightful suggestions will greatly contribute to enhancing Jailbreak-AudioBench, thereby facilitating the advancement of progress in audio jailbreak research.

---

### Note · Authors · 2025-08-13

Dear Area Chair,

We sincerely thank you and all Reviewers (R) for your efforts in the review process and for recognizing the significance and novelty of our work on Jailbreak-AudioBench, as reflected in comments such as “**Overall, it is a good paper**” (R-KQbp), “**the paper is well-structured that enhances the paper’s readability and impact**” (R-LwpR), “**a highly novel and important perspective by focusing on the safety of Audio Language Models**” (R-11tt), and “**I think this is a strong paper that will help pave the way for research on jailbreaking LLM in the audio domain**” (R-hifu).

Moreover, we genuinely appreciate the valuable suggestions provided by all Reviewers. Below, we provide a summary of the discussions.

- **Related Work Integration (R-KQbp)**: R-KQbp recommended incorporating recent audio deepfake research to contextualize our work better. We integrated the suggested references to strengthen the paper's positioning in the audio security landscape.

- **Model and Dataset Coverage (R-11tt & R-LwpR)**: R-11tt recommended evaluating proprietary models like Gemini. R-LwpR suggested incorporating real human recordings. We conducted experiments on Gemini-2.5-Flash with the Attack Success Rate (ASR) increasing from **8.1% to 49.4%**. We also launched a "Voice Recording" activity for collecting real human speech samples.

- **Statistical Reporting (R-LwpR)**: R-LwpR recommended multiple random seeds and statistical reporting for experiments. We conducted multi-seed experiments on subsets of our dataset and provided ASR results with means ± standard deviations.

- **Human Evaluation (R-11tt)**: R-11tt suggested reporting details about human evaluation procedures. We provided human evaluation details, including the number of annotators, annotation procedures, and inter-annotator agreement metrics with Cohen's κ score.

- **Automated Evaluation (R-11tt)**: R-11tt pointed out that harmfulness assessment can be enhanced by new automated evaluation methodologies. We supplemented our Llama Guard 3 assessments with GPT-4o evaluations for whether jailbreaks were successful.

- **Dataset Categorization (R-hifu & 11tt)**: R-hifu recommended clearly explaining the criteria for categorizing audio as explicit or implicit, while R-11tt suggested including details of sample distribution. We clarified our categorization process and provided detailed sample counts for each jailbreak task.

Sincerely,

The Authors of Submission 557

---

### Decision · Program_Chairs · 2025-09-18

**Decision:**

Accept (poster)

**Comment:**

This paper introduces Jailbreak-AudioBench, a benchmark for evaluating the robustness of auto-language models against adversarial prompts. The work addresses a timely and underexplored problem in evaluating multi-modal (audioLLM) safety. All reviewers found the benchmark constructed reasonably and the finding insightful, although there are also noted limitations such as the reliance on TTS-generated attacks and the lack of evaluation on real-world scenarios. The rebuttal clarified these choices and outlined possibilities for future extensions. While there is still room for improvement such as generalizability and attach coverage, the AC finds the contribution is quite relevant and worthy for acceptance.